# The Pivotal Role of the Dysregulation of Cholesterol Homeostasis in Cancer: Implications for Therapeutic Targets

**DOI:** 10.3390/cancers12061410

**Published:** 2020-05-29

**Authors:** Etienne Ho Kit Mok, Terence Kin Wah Lee

**Affiliations:** 1Department of Applied Biology and Chemical Technology, The Hong Kong Polytechnic University, Hong Kong 999077, China; ho-kit.mok@connect.polyu.hk; 2State Key Laboratory of Chemical Biology and Drug Discovery, The Hong Kong Polytechnic University, Hong Kong 999077, China

**Keywords:** cancer, cancer stem cells, cholesterol homeostasis, cholesterol esters, cholesterol-lowering drugs, oxysterols, immunotherapy, lipoproteins, statins

## Abstract

Cholesterol plays an important role in cellular homeostasis by maintaining the rigidity of cell membranes, providing a medium for signaling transduction, and being converted into other vital macromolecules, such as sterol hormones and bile acids. Epidemiological studies have shown the correlation between cholesterol content and cancer incidence worldwide. Accumulating evidence has shown the emerging roles of the dysregulation of cholesterol metabolism in cancer development. More specifically, recent reports have shown the distinct role of cholesterol in the suppression of immune cells, regulation of cell survival, and modulation of cancer stem cells in cancer. Here, we provide a comprehensive review of the epidemiological analysis, functional roles, and mechanistic action of cholesterol homeostasis in regard to its contribution to cancer development. Based on the existing data, cholesterol homeostasis is identified to be a new key player in cancer pathogenesis. Lastly, we also discuss the therapeutic implications of natural compounds and cholesterol-lowering drugs in cancer prevention and treatment. In conclusion, intervention in cholesterol metabolism may offer a new therapeutic avenue for cancer treatment.

## 1. Introduction

Cholesterol is gaining increasing attention in cancer research due to its targetable therapeutic implications in both the prevention and treatment of cancer. However, the role of cholesterol in tumorigenicity remains controversial [1]. Researchers have reported a distinctively contradictory role of cholesterol in cancer development, showing that the correlation of cholesterol in carcinogenicity can be cancer-type specific [2]. High cholesterol or hypercholesteremia has positive correlations in breast and prostate cancers [3,4], while some prospective cohort studies show an inverse association [5,6]. Therefore, this review aims to discuss the current understanding of cholesterol homeostasis, to summarize the key findings of recent pre-clinical and clinical studies investigating cholesterol metabolism in cancer, and to provide up-to-date therapeutic implications of natural compounds and cholesterol-lowering drugs in cancer treatment.

## 2. Physiological and Functional Roles of Cholesterol Homeostasis

As a subtype of lipid, cholesterol exists in every type of mammalian cell, ranging from fundamental components of cell membranes by maintaining integrity and stability to precursors of different forms of vital sterol compounds, like vitamins and hormones. Akin to other important molecules, cellular cholesterol levels are tightly regulated through metabolic processes, namely de novo biosynthesis, intake, export, and esterification of excess free cholesterol [7].

The de novo cholesterol biosynthesis, or the mevalonate pathway in some contexts (Figure 1a), consists of more than 20 enzymatic steps. It all starts from combining three acetyl-CoA molecules to form one 3-hydroxy-3-methylglutaryl coenzyme A (HMG-CoA). Under the first rate-limiting catalytic enzyme, HMG-CoA reductase (HMGCR) converts HMG-CoA into mevalonate, which is then transferred to farnesyl pyrophosphate (FPP), squalene, and finally cholesterol through a series of enzymatic reactions. Noteworthy, FPP, apart from transferring into downstream sterols and all other nonsterol isoprenoids, is capable of converting into geranylgeranyl pyrophosphate (GGPP), which are both important effectors in protein prenylation [8,9].

Meanwhile, the mevalonate pathway is supervised by a master transcriptional regulatory protein called sterol-regulatory element binding protein 2 (SREBP2) in a negative feedback loop [10]. SREBP2 is synthesized in a premature format in the endoplasmic reticulum (ER) membrane [11]. The maturation of SREBP2 requires a two-step proteolytic cascade that occurs in the Golgi apparatus, where site 1 protease (S1P) and S2P act consecutively to release the N-terminal fragment of SREBP2 [11,12]. This N-terminal fragment, or nuclear SREBP2 (nSREBP2), enters the nucleus and binds to the genes containing sterol regulatory elements at the promoter region, subsequently enhancing their transcriptional levels [13]. Those genes are therefore involved in cholesterol biosynthesis, such as HMGCR, low density lipoprotein receptor (LDLR), and squalene synthase [13]. The activation of SREBP2 occurs only when the intracellular cholesterol level is low so that the SREBP cleavage-activating protein (SCAP), another ER-anchored protein, is freed from cholesterol and insulin-induced gene protein 1 (INSIG1) [14,15]. The detachment of INSIG1 induces a closed conformational change of SCAP and allows its binding to COPII-coated vesicles [16]. The SCAP-SREBP2 complex is transported to the Golgi apparatus together with the COPII vesicles for SREBP2 activation. However, when the intracellular cholesterol level is high enough to bind to SCAP, which sequentially recruits INSIG1, the INSIG1 attachment relaxes the conformation of SCAP, thus prohibiting the complex from binding to COPII vesicles [16]. The transcriptional activity of nSREBP2 can also be increased by the master regulator of anabolic reactions, mammalian target of rapamycin complex 1 (mTORC1), via inhibition of nuclear entry of lipin1, which downregulates nSREBP2 [17]. Apart from that, the mevalonate pathway, which is an energetically expensive metabolic process, can also be regulated through rate-limiting enzymes. HMGCR can be phosphorylated by 5′ adenosine monophosphate-activated protein kinase (AMPK) to abolish its activity when intracellular ATP levels are low [18]. Recently, squalene epoxidase (SQLE) has been considered as another rate-limiting enzyme in this pathway, in which it converts squalene into squalene epoxide [19]. The E3 ubiquitin ligase MARCH6 is recruited to degrade squalene epoxidase when excess cholesterol is present [19].

When de novo biosynthesis remains the main source of intracellular cholesterol, most cells acquire cholesterol from low density lipoprotein (LDL) in the circulatory system via LDLR-mediated endocytosis [20]. Free cholesterol is then dissociated from LDL when lysosome is digested. Yet, proprotein-convertase-subtilisin-kexin type-9 (PCSK9) induces lysosomal degradation to LDLR [21]. The very-low-density lipoproteins, the precursors of LDL, are composed in liver, where the dietary cholesterol is transported for compartmentation. In contrast to LDLR-mediated endocytosis, enterocytes in the intestinal lumen absorb dietary cholesterol via Niemann–Pick type C1-like 1 protein (NPC1L1) through a clathrin-dependent pathway [22]. The upregulation of NPC1L1 contributes to cardiovascular diseases and symptomatic gallstone diseases [23].

When cholesterol has served its intracellular purposes, the excess cholesterol is exported via ATP-binding cassette (ABC) subfamily A member 1 (ABCA1) or ABC subfamily G member 1 (ABCG1) to lipid-poor apolipoprotein A-I (ApoA-I) and generates high-density lipoproteins (HDLs) that are transported back to the liver [24,25,26,27]. The transcriptional level of ABCA1 is upregulated by nuclear liver X receptor (LXR) when the intracellular cholesterol level is high [27]. Surplus cholesterol can also be esterified by acyl-CoA:cholesteryl acyltransferase 1 (ACAT1) into cholesteryl esters (CEs), which are a less toxic form and can be stored as lipid droplets or for further processing into lipoproteins [28].

The discovery of microRNAs (miRNAs), a class of non-coding RNAs, has added complexity into cholesterol homeostasis through regulation of different key components in the system [29]. miR-33a, an embedded intronic microRNAs, is located within SREBP2 gene [30]. In a low sterol condition, akin to SREBP2, miR-33a is transcribed up to 2- to 3-fold higher to regulate cholesterol export and HDL metabolism gene by targeting ABCA1 for post-transcriptional repression [30]. On the other hand, miR-223 controls cholesterol level by inhibiting the synthesis and enhancing the cholesterol efflux by elevating the expression of ABCA1 [31]. miRNA-122 is highly expressed in hepatocytes, accounting for 70% of all liver miRNA [32]. Inhibition of miRNA-122 substantially suppresses total plasma cholesterol [32]. However, the direct target of miRNA-122 is yet to be elucidated [32]. Apart from these miRNAs, miR-27a has been shown to specifically interact with HMGCR 3′ untranslated region to inversely regulate HMGCR expression by posttranslational inhibition followed by mRNA degradation [33]. Strikingly, the application of genome-wide association studies has allowed the discovery of more miRNAs in abnormal levels of cholesterol-lipoprotein circulation, such as LDLR and ABCA1 [34]. These mRNAs include miR-128-1, miR-148a, miR-130b, and miR-301b [34]. Taken all together, these findings have suggested the potential involvement of miRNAs in regulation of cholesterol metabolism, and they may contribute to an abnormal cholesterol level if left unregulated.

## 3. The Relationship between Cholesterol and Cancer Incidence

As an essential macromolecule in metabolism, cholesterol has been suspected to play an important role in inducing cancer. Since the 1980s, such a relationship has been extensively monitored and examined in different clinical cohort studies. However, the results have indicated that the relationship between cholesterol and cancer is type- and stage-specific, both to the tumor originating site and the form of lipoproteins being examined. Cholesterol circulates in the body mainly in two different forms: LDL or HDL. Researchers have examined them separately or inclusively as total cholesterol (TC) to determine their tumorigenicity effect (Table 1).

LDL cholesterol (LDL-C) level has been suggested to be a prognostic factor of breast cancer progression at diagnosis [49]. A prospective study on 244 women with operable breast cancer in Portugal showed that patients with LDL-C levels as high as 117 mg/dL had poor prognosis due to the higher proliferative rate, histological stage, and more advanced clinical stage [49]. Meanwhile, patients with LDL-C levels above 144 mg/dL can suffer from lympho-vascular invasion as well as lymph node metastasis [49]. Yet, in two meta-analyses involving over 1 million patients each, LDL-C showed no association with breast cancer risk irrelevant to menopause in women [50,51]. On the other side of the world, in a large population-based case-control study conducted in Shanghai, China, researchers showed the relationship of LDL-C in biliary tract cancers [43]. Patients contracting bile duct cancer had significantly higher LDL-C levels than control patients, while those patients who suffered from gallbladder cancer had lower LDL-C. However, there was no significant difference in LDL-C level in patients contracting carcinoma of the ampulla of Vater, a small region connecting the duodenum, bile duct, and pancreatic duct, compared with the control group [43]. Moreover, LDL-C was positively associated with liver metastases in colorectal cancer patients [52].

In contrast, HDL cholesterol (HDL-C) may have a clearer effect on reducing the cancer risk. A prospective follow-up study of participants who were enrolled in the ATBC Cancer Prevention Study showed a strong inverse association between HDL-C and non-Hodgkin lymphoma (NHL). The researcher claimed that the risk of NHL was reduced by 15% for each 5 mg/dL increase in HDL-C level [39]. Participants with HDL-C levels above 55 mg/dL had over 60% lower risk of developing NHL [39]. A similar situation was observed in biliary tract cancers in the study mentioned above. Patients with a high HDL-C level (>40 mg/dL) had 11.6- and 16.8-fold lower risks of gallbladder and bile duct cancers than the patients with low HDL-C levels (<30 mg/dL) [43]. Additionally, a modest association of HDL-C with <50 mg/dL could increase the breast cancer risk among premenopausal women in a prospective study examining over 7000 patients [53].

Taking both circulating forms into account and measuring them as the TSC level provides a complementary, yet more elusive, picture to examine the relationship of cholesterol and cancer risk in clinical studies. In an all-inclusive prospective study of nearly 1.2 million Korean participants, the investigators examined several common types of cancers, including stomach, liver, pancreas, lung, prostate, or colon cancers [41]. After adjusting for body mass index, alcohol consumption, fasting glucose, hypertension, smoking, and physical activity, inverse associations were observed between all-cancer incidence with total cholesterol in men (Hazard Ratio (HR), 0.84; 95% Confidence Interval (CI), 0.81 to 0.86) and in women (HR, 0.91; 95% CI, 0.87 to 0.95) [41]. A similar result was observed in another large prospective study consisting of seven cohorts from Norway, Austria, and Sweden, including nearly 600,000 participants [37]. Inverse associations were demonstrated between total cholesterol level and all cancer risk in men (HR, 0.94; 95% CI, 0.88 to 1.00) and in women (HR, 0.86; 95% CI, 0.79 to 0.93) [37]. In some individual cancer types, positive correlations could be established in both sexes, such as prostate, colon, pancreatic, and breast cancer [37]. Particularly in prostate cancer, men with higher cholesterol levels were at greater risk of developing a higher clinical stage of prostate cancer [36].

## 4. Critical Oncogenic Pathways in Cholesterol Homeostasis

Dysregulation of key molecules in cholesterol homeostasis or cholesterol itself has not only been associated with several well-known oncogenic pathways, but also related to inflammasome- and miRNAs-mediated cancer development (Figure 1b). By understanding the interplay between these parties, more effective drug interventions can be developed.

In the mevalonate pathway, production of FPP and GGPP could induce onco-protein prenylation, which is involved in the activation of several oncoproteins, such as Ras GTPases [8,9,54]. Meanwhile, proprotein-convertase-subtilisin-kexin type-9 (PCSK9) induces lysosomal degradation to LDLR [21]. However, the overexpression of PCSK9 contributes to hypercholesterolemia and sequentially correlates with hepatocellular carcinoma development [55]. Hyperactivity of LXR, another key player in cholesterol homeostasis induced by its agonists, has been shown to exert an anti-proliferative effect in gastric cancer cells [56]. Yet, though CEs serve as a cholesterol reservoir, the accumulation of CEs or overexpression of ACAT1 have supported a pro-tumor role. In hepatocellular carcinoma, ACAT1 elevation is identified by proteomic and phospho-proteomic analyses [57]. In xenograft models of glioblastoma, ACAT1 ablation has reduced the tumor progression [58]. CP-113818, the ACAT1 inhibitor, suppresses the migration capacity of breast cancer cells [59]. Furthermore, inhibition of ACAT1 has also been shown to decrease prostate cancer progression [60].

One of the most intensively studied oncogenes, *TP53* gene mutation, arising from deletion or truncation, aggressively promotes tumor survival, invasion, migration, metastasis, and chemoresistance in many cancers [61]. With functional p53 protein, SREBP2 activity is suppressed due to upregulation of ABCA1, hence reducing the transcriptional levels of enzymes in the mevalonate pathway [62]. However, with respect to breast cancer, p53 disrupts the acinar morphogenesis, or tissue architecture, of breast cells, aided by the upregulated expression of the cholesterol biosynthesis pathway. By harvesting tumor-derived mutants of p53 in an organoid culture system, through Ingenuity Pathway Analysis and Gene Ontology Analysis, cholesterol biosynthesis was shown to be the most overrepresented regarding p53 downregulation [63]. A rescue experiment supplementing the essential intermediate metabolites in the mevalonate pathway could significantly inhibit the disordered phenotype caused by silenced p53 in breast cancer cells in 3D culture [63]. Moreover, TP53-mediated SREBP2 cholesterol synthesis can also enhance the prenylation of Rho GTPases, which, in turn, enhance the proliferation and self-renewal of breast cancer cells [54,63]. Conversely, simvastatin strikingly decreased cancer cell growth, increased cell death, reduced invasiveness, and mimicked the mutant p53 depletion in terms of morphological changes [63].

Another example is the phosphoinositide 3-kinase (PI3K)/protein kinase B (Akt)/mTORC1/SREBP signaling axis, which induces overall cell growth. While the PI3K and Akt signaling pathway is responsible for the accumulation of mass, the inhibition of mTORC1 can attenuate Akt-dependent lipogenesis and eventually cause reductions in cell size in vitro and in vivo [64,65]. This finding was further elaborated. The abnormal activation of the PI3K/Akt signaling pathway maintains a high intracellular level of cholesterol via mTORC1 in inhibiting the ABCA1 efflux activity and via SREBP overexpression, resulting in LDLR-dependent cholesterol import [66]. Moreover, a high intracellular cholesterol level could further drive mTORC1 recruitment and activation in lysosomes via lysosomal transmembrane protein SLC38A9 [67]. Meanwhile, in prostate cancer, the loss-of-function phosphatase and tensin homolog (PTEN) can activate the PI3K/Akt pathway and lead to the accumulation of CEs as a result of excess intracellular cholesterol levels after upregulation of LDLR-mediated cholesterol influx [68]. A similar result was also seen in nonalcoholic fatty acid induced hepatocellular carcinoma. The overexpression of SQLE can suppress PTEN activity and subsequently induce the accumulation of CEs through Akt signaling [69]. Collectively, such an alteration is related to cell proliferation, tumor formation, and cancer aggressiveness in terms of invasion and metastasis in cancers.

Cholesterol and its oxygenated derivatives have shown strong affinity to G protein-coupled receptor (GPCR), i.e., Smoothened receptor (SMO), which activates the sonic hedgehog (SHH) pathway [70]. The SHH pathway is considered an oncogenic signaling cascade, as it is capable of promoting cell cycle progression and stem cell proliferation through increased activity of GLI1 and subsequent activation of hedgehog targeted genes, therefore enhancing tumor formation [71]. Inhibition of cholesterol synthesis by statins can successfully arrest SHH signaling in medulloblastoma cells and fibroblasts, thus attenuating the proliferation of tumors [72].

Inflammation is an immune response to endogenous danger signals which helps to combat different stresses [73]. The causal relationship between chronic inflammation and cancer is widely established nowadays [74]. Inflammasomes, the large intracellular multi-protein signaling complexes, are formed under inflammation which help to activate inflammatory protease caspase-1, pro-inflammatory cytokines interleukin (IL)-1β and IL-18 [74]. Nod-like receptor protein 3 (NLRP3) is one of the most well studied inflammasomes among the families and its dysregulation is associated to cancer development. The uncontrolled formation of NLRP3, arising from different cellular challenges such as presentation of lipopolysaccharides, viruses, or abnormal ion fluxes, induced IL-1β and IL-18 productions, resulting in development of various cancer types, including head-and-neck squamous cell carcinoma [75], oral squamous cell carcinoma [76], and breast cancer [77]. In colorectal cancer, cholesterol promoted colon carcinogenesis through activating the NLRP3 inflammasome and suppression of AMPKα in macrophages, resulting in significant increase of mitochondrial reactive oxygen species, which in turn enhanced the NLRP3 inflammasome activity [78]. A similar positive feedback loop was observed in hepatocellular carcinoma [79]. The enhanced production and accumulation of cholesterol in liver cancer cells activated NF-κB signaling, which could promote the overall cholesterol production via activating SREBP2, HMGCR, and LDLR [79]. Yet, in other cells like endothelial cells, SREBP2 is an important mediator for NLRP3 inflammasome activation and amplification via SREBP2-TIFA and SREBP2-NOX2 cascade [80,81], further strengthening the relationship between cholesterol homeostasis and inflammation.

Accumulative evidence has demonstrated the crucial role of miRNAs in cancer development [82,83]. MiR-122 was found to regulate cholesterol homeostasis, and its overexpression is required for hepatitis C virus propagation and accumulation through binding to the 5′ UTR of the hepatitis C virus genome [84,85]. Meanwhile, miR-183 promoted proliferation and anti-apoptotic properties in colon cancer cells, through retaining a high level of intracellular cholesterol via direct degradation of ABCA1 mRNA [86]. Similarly, miR-27 also exerts anti-apoptotic function in cancer cells by blocking cholesterol efflux or targeting ABCA1 [87]. Furthermore, MYC exerts its oncogenic effects in part by altering mevalonate metabolism in glioma cells via induction of miR-33b [88]. However, on the other hand, miRNAs could act as onco-suppressors. Inhibition of miR-612 induced HADHA overexpression which in turn modified cholesterol biosynthesis via SREBP2/HMGCR cascade, eventually leading to invadopodia formation and metastasis of HCC [89]. Lastly, miR-33a has shown to be suppressed in tumors derived from lung cancer [90], breast cancer [91], and colorectal cancer [92]. Particularly in colorectal cancer, cholesterol can regulate cancer development, cell cycle progression, and anti-apoptosis via miR-33a-PIM3 signaling pathway [92].

## 5. The Role of Cholesterol Metabolism in the Regulation of Cancer Stem Cells

Cancer stem cells (CSCs), or tumor-initiating cells (T-ICs), have been proposed to play important roles in tumor initiation, recurrence, and chemoresistance, in which dysregulated cholesterol metabolism is shown to be involved [93]. Though as a small subset inside cancer cells, a growing body of evidence of the utility of T-ICs or cells showing stem-like characteristics has tried to explain the failure of current conventional chemotherapy in which patients or cancer cells can acquire resistance to chemotherapeutic drugs after certain periods of drug administration, eventually leading to tumor relapse [94]. Therefore, isolation of T-ICs and identification of their critical signaling pathways would bear clinical significance in light of registering new targeted therapies against virtually all types of cancer.

By transforming immortalized human fibroblasts into cells bearing CSC phenotypes, cells overexpressing some of the stemness regulator genes, such as sex determining region Y-box 2, octamer-binding transcription factor 4, and homeobox protein, can form tumorspheres in an anchorage-independent manner and develop tumors in immunodeficient mice [95]. Of note, a global genome expression microarray has recognized atypical metabolic pathways when comparing the sphere-forming cells against their differentiated counterparts. The sterol biosynthetic process, or cholesterol biosynthetic process, has ranked in the top five out of 15 biological processes, showing the abnormal exploitation of cholesterol in tumor formation, particularly in supporting the growth of CSCs [95].

Likewise, in cancer cell lines, cholesterol biosynthesis has intimate linkage to CSC population proliferation. In colorectal cancer cells, LDL, which is the main carrier form of cholesterol in blood vessels, was shown to regulate stemness in vitro by promoting stem-like characteristics, including spheroid formation capacity, stemness-regulating genes and migration capacity [52]. Interestingly, LDL enhanced colorectal cancer progression via the MAPK pathway, which was associated with cell proliferation and differentiation. Apart from the elevation of cholesterol intake receptor, key enzymes involved in cholesterol de novo biosynthesis are also shown to be altered in every aspect. In glioblastoma, by running and comparing the RNA sequencing of patient-derived glioblastoma sphere cells and their differentiated counterparts, the super-pathway of cholesterol biosynthesis was shown to be predominantly upregulated [96]. Among those gene lists, enzymes involved in the mevalonate pathway, which mainly synthesize sterols as end-products, including farnesyl-diphosphate farnesyltransferase 1, farnesyl diphosphate synthase (FDPS), and 3-hydroxy-3-methyglutaryl-CoA synthase 1, were highly upregulated when compared to differentiated glioblastoma cells [96]. Addition of inhibitors (alendronate and zoledronate against FDPS) has rescued the tumor progression effects [96]. Similarly, breast cancer stem cells were also shown to be tightly regulated by this mevalonate metabolism pathway [97]. Intriguingly, metformin, an anti-diabetes drug, has prohibited cancer cell growth by lowering cellular cholesterol content and the stemness properties in breast cancer cells [98] and also by reducing the numbers of tumor-initiating epithelia cell adhesion molecule (EpCAM)^+^ hepatocellular carcinoma cells [99]. Meanwhile, excess cholesterol can inactive lysophosphatidylcholine acyltransferase 3 (Lpcat3), which is responsible for polyunsaturated phospholipid synthesis and drives stem cell proliferation in intestinal cancer in vivo and ex vivo [100]. Alternatively, inhibition of Lpcat3 or overexpression of master regulator of mevalonate pathway, SREBP2, markedly promotes intestinal tumor formation in tumor suppressor gene adenomatous polyposis coli (Apc) multiple intestinal neoplasia (Min), or *Apc ^min^*-induced tumor mice [100].

## 6. The Role of Cholesterol Metabolism in Immune Cells

As an important element in the mammalian phospholipid bilayer membrane, cholesterol helps to maintain the membrane’s rigidity and to provide a medium for proper cellular signal transmission, particularly in lipid rafts [7,101]. To achieve high levels of proliferation and activation, the propagation of the cell membrane is critical not only for cancer cells but also for the maturation of immune cells in response to adverse stress. Furthermore, the receptor relocation or co-localization in the immune cells is also critical for proper activation, in which cholesterol or its derivatives participate [102]. Regarding cancer scenarios, the relationship of cholesterol to immune cells is again cell-type specific, as we will discuss in this section.

T cell proliferation and activation require massive amounts of energy to support various forms of biosynthesis, and it is reported that fatty acid biosynthesis as well as cholesterol biosynthesis are highly upregulated in T cells [103]. When T cells receive an activation signal, SREBP2 maturation is granted, while LXR is inactivated [104]. Following LXR inactivation, the cholesterol efflux transporter ABCG1 is also suppressed [104,105]. Another protein inhibiting the cholesterol efflux transporter, mTOR, has also been shown to regulate CD8^+^ T cell differentiation through regulating cholesterol metabolism [106]. The overall intracellular cholesterol retention is beneficial for T cell activation, and SREBP is demonstrated to show an important role in CD8^+^ T cell activation and proliferation [107]. Alteration of cholesterol concentration can also result in insufficient composition of lipid rafts, which allow the interaction of membrane-associated proteins. Under the administration of the lipid raft disruption mediator Miltefosine, T cell proliferation is retarded by over half when compared to the control [108]. Cholesterol itself can also be regarded as a signaling molecule in the T cell community. The administration of squalene, a precursor of cholesterol, increases the population of CD4^+^ T cells and predisposes T cells in response to inflammatory action [109]. In contrast, upon removal of cholesterol from either growth medium or the mouse diet, total T cell activation and proliferation are retarded [110]. Meanwhile, the bloodstream levels of LDL or HDL could compromise initial T cell development, as total cholesterol is reduced [111]. The inhibition of ACAT1, which esterifies cholesterol into CEs, activates CD8^+^ T cell as the total plasma cholesterol level increases [112]. Oxysterols, such as 27-hydroxycholesterol, have been found to attract γδ T cells but exhaust CD8^+^ T cells, eventually prompting breast cancer metastasis [113]. Therefore, the accumulation of cholesterol can facilitate nanoclustering in T cells, ultimately promoting the antigen-presenting capacity and upregulation of cholesterol synthesis and influx [114]. Akin to the controversial effects of cholesterol in cancer, cholesterol has been shown to negatively regulate T cell activities. In the tumor microenvironment, high cholesterol levels can lead to CD8+ T cell exhaustion while inducing immune checkpoints, such as programmed cell death protein 1 (PD-1), natural killer cell receptor 2B4 (CD244), T-cell immunoglobulin and mucin-domain containing-3 (TIM-3), and lymphocyte-activation gene 3 (LAG-3), via the ER stress sensor XBP1 signaling cascade [115]. By suppressing the XBP1 transcriptional capacity or reducing cholesterol content in the microenvironment, the anti-tumor activity of CD8^+^ T cells can be restored [115]. In addition, CD8^+^ T cells are capable of differentiating into different subsets with various cytokine expression profiles. Among those subsets, interleukin-9-secreting T (Tc9) cells are reported to exert stronger antitumor effects when compared to Tc1 cells [116]. However, cholesterol has been shown to attenuate IL-9 expression via activating the LXR signaling cascade and inhibiting Tc9 cell activity in antitumor responses [116].

Apart from T cells, other tumor-infiltrating cells have also been shown to be regulated by cholesterol or its oxidative derivatives. Neutrophils are attracted by hypoxia-inducible factor-1α (HIF1α) under the elevation of 24-hydroxycholesterol, ultimately inducing angiogenesis [117]. 25-hydroxcholesterol has been shown to advance gastric cancer metastasis in vitro by enhancing matrix metallopeptidase expression while interacting with GPCRs to trigger macrophages [118]. Cholesterol is also found to accumulate in natural killer cells and to aid in lipid raft formation and immune signaling activation [119]. The overall maturation of natural killer cells will eventually retard mouse hepatoma cell development, thus demonstrating a strategy in combating hepatocellular carcinoma [119]. The effect of oxysterol in an immunosuppressive role can be restored in dendritic cells by expression of sulfotransferase 2B1b (SULT2B1b), which converts oxysterols into impotent sulfated oxysterol [120]. Moreover, the antigen presentation efficiency of dendritic cells can be enhanced by utilizing the natural influx of cholesterol into the cells [120]. Cholesterol-modified antimicrobial peptide (AMP) DP7 (DP7-C) can efficiently deliver various antigen peptides into dendritic cells via clathrin- and caveolin-dependent pathways, thus inducing dendritic cell maturation [121]. This novel antigen-presenting technique can be utilized as a personalized cancer immunotherapy that has demonstrated excellent antitumor effects in mouse tumor models.

## 7. Effect of Anti-Cancer Drugs and Natural Compounds on Cholesterol Homeostasis Pathway in Cancer Cells

Accumulating evidence suggests that anticancer drugs may exert their anti-proliferative activities at least in part by reducing cholesterol content/biosynthesis. For instance, it has been demonstrated that doxorubicin induced cancer cell death by decreasing HMGCR expression and reducing cholesterol levels, which was mediated by downregulation of HMGCR via inhibition of EGFR/Src pathway [122]. Other studies indicated that tamoxifen modulates cholesterol metabolism in breast cancer cells [123]. In addition, a recent article highlighted that BRD4 inhibitor JQ1 severely impacts the expression of proteins involved in cholesterol metabolism, thus leading to a strong decrease of cholesterol content in the human liver cancer cell line HepG2 [124]. Interestingly, the same report showed that administration of cholesterol counteracts the anti-proliferative effect induced by JQ1 in hepatocellular carcinoma cells, and the acquisition of JQ1-resistance is accompanied by a compensatory upregulation of proteins belonging to cholesterol homeostasis [124]. All these solid data showed the critical role of cholesterol homeostasis in tumor cell survival in response to various anti-cancer drug treatments.

Apart from anti-cancer drugs, many natural compounds also exhibit a therapeutic role in cancer prevention and therapy. Many of them, including terpenoids, green tea, garlic extract, and curcumin, were found to target cholesterol homeostasis in cancer cells. Isoprenoids, also known as terpenoids, are a class of naturally occurring phytochemicals found in fruits, vegetables, and unrefined cereal grains. Several isoprenoids such as δ-, γ-, and α-tocotrienol [125], β-ionone [126], geranylgeraniol [127], and geraniol [128], were shown to suppress the growth of tumor cells by inhibiting the transcription and activity of HMGCR in various cancer types. In a large-scale compound screening, ursolic acid, a pentacyclic terpenoid, was also found to exert anti-cancer effects in hepatocellular carcinoma cells via suppression of cholesterol biosynthesis [129]. Green tea polyphenol (EGCG) was also widely reported to exert anti-cancer role in various cancers. EGCG modulates cholesterol metabolism by increasing the efflux of cholesterol and directly inhibiting HMGCR [130,131]. The cholesterol-lowering effect of EGCG was further confirmed in human clinical studies [132]. Garlic extract was reported to decrease cholesterol biosynthesis by inhibiting sterol 4alpha-methyl oxidase [133]. In addition, several garlic-derived organosulfur compounds, including S-allylcysteine and ajoene, have been found to inhibit HMGCR activity [134]. Curcumin has a long history of use as an anti-inflammatory agent. The active component of curcumin was found to induce cell death of tumor cells. Recently, curcumin was found to suppress cholesterol biosynthesis superpathway via targeting squalene monooxygenase, which was found to complement the effect of statin in cancer therapy [135]. In addition, curcumin was able to suppress cholesterol uptake in colon cancer cells by downregulation of NPC1L1 expression [136]. Collectively, natural compounds can be used to regulate cholesterol homeostasis not as a primary cancer therapy but also as an adjuvant to complement current molecular therapies.

## 8. Molecular Targeted Drugs Targeting Cholesterol Homeostasis Pathway for Cancer Prevention and Treatment

Considering the elevated cholesterol levels in different types of cancer, as discussed in previous chapters, no matter if they are the result of the overexpression of cholesterol biosynthetic genes, enhancement of the cholesterol import mechanism or suppression of cholesterol export activity, abnormal cholesterol content can eventually activate oncogenic pathways, or their derivatives can exert immunosuppressive roles. Therefore, it is practical to apply cholesterol-lowering drugs to prevent cancer incidence and to treat cancer.

Regarding the de novo biosynthesis of cholesterol, the process involves more than 20 enzymes that are potential candidates of drug intervention to regulate the overall cholesterol biosynthesis. Among those proteins, HMGCR has been well established as a rate-limiting catalytic enzyme in the mevalonate pathway, converting HMG-CoA to mevalonic acid. Given its important characteristics, statins, first marketed in 1987, have been used to inhibit HMGCR by functioning as a HMG-CoA analogue, eventually decreasing the cholesterol content [137]. Different subclasses of statins have been synthesized, arising from distinctive lipophilicity and the capacity to cross the blood–brain barrier. Since it is beyond the scope of this review, the differential properties can be read in other papers [138,139,140]. Statins were first used for treating atherosclerosis, cardiovascular diseases, and liver diseases, which arise from the excess deposition of cholesterol [141,142]. However, with accumulating clinical evidence, statins have been considered as an anti-cancer drug in recent decades [143]. In prostate cancer, statin use could effectively reduce the mortality and reduce the risk of prostate-specific antigen (PSA) recurrence in a dosage-dependent manner after radical prostatectomy [144]. Patients with metastatic renal cell carcinoma benefited from statin use in terms of improved overall survival (25.6 versus 18.9 months) [145]. The clinical data are further supported by several epidemiologic studies. The incidence rates of cancers, for example, liver, gastric, colorectal, pancreatic, and prostate cancers, are reduced under the administration of statins (Table 1). Meanwhile, laboratory experiments are also able to show that the use of statins could decrease proliferation and viability of human cancer cell lines [146,147,148].

Other enzymes involved in the mevalonate pathway can also be targeted (Table 2). Bisphosphonates have been used to inhibit FPP synthase in converting mevalonate into farnesyl diphosphate [149]. Lapaquistat is used to inhibit squalene synthase [150], while Lamisil is used to attenuate SQLE, which is considered to be an oncogene [151]. Zaragozic acids, which also inhibit the production of squalene, show inhibitory effects in both lung carcinoma and lymphoma growth [152]. Another potential inhibitor to the oxidosqualene cyclase (OSC), Ro 48-8071, effectively reduces the progression and metastasis of pancreatic and colorectal cancers via prohibiting the production of lanosterol, thus limiting cell proliferation and migration [153].

Apart from inhibiting the cholesterol synthesis inside the body, the intake from dietary cholesterol can be intervened by taking Ezetimibe, which disrupts the NPC1L1 protein on enterocytes and lowers LDL-C [163,164]. This has been shown to inhibit the tumor angiogenesis in prostate tumors and, hence, progression [163]. Previously, we have shown that CEs assume a pro-tumor position. Therefore, inhibition of the production of CEs shows promising anti-tumor effect. For example, the administration of the ACAT1 inhibitor avasimibe suppresses CE production and even restores the imatinib sensitivity in a myelogenous leukemia cell line and, hence, retards the growth [165]. A similar effect of ACAT1 inhibition in tumor suppression has also been demonstrated in prostate and triple-negative breast cancer cells [166,167]. Methyl-β-cyclodextrin [MβCD], a cholesterol depletion chemical, is used to disrupt the lipid rafts, which are important segments in the cell membrane for proper signaling transduction and oncoprotein embedment [162]. MβCD has been shown to induce apoptosis in breast cancer cells via activating the pro-apoptotic caspase-3 signaling cascade [168]. Given that MβCD disrupts the membrane integrity, it can synergize the efficiency of tamoxifen, as the drug can easily pass through the membrane [169,170,171].

Finally, a combination treatment using conventional anti-cancer drugs and drugs targeting cholesterol metabolism proposes a promising result in treating cancers. For example, in sterol hormone-related cancers, the administration of statins sensitizes anti-hormonal drugs in breast and prostate cancers [22,172,173]. Ro 48-8071, an inhibitor of OSC, has improved tissue perfusion and thus synergizes the 5-fluoroouracil anti-tumoral effect in human colon carcinoma [153]. Avasimibe, the inhibitor of cholesterol esterification, has been well documented with chemotherapeutic drugs such as doxorubicin in inducing apoptosis in a tumor model [174]. Avasimibe is also considered an immunotherapeutic, as it can boost adaptive anti-tumor immunity in head-and-neck cancer cells when combined with a dendritic cell vaccine, or it controls melanoma progression when combined with anti-PD-1 immunotherapy [175].

## 9. Conclusions

This review has discussed the important role of cholesterol metabolism in cancer development. Elevated cholesterol contents are observed in different cancers and in the reprogramming of cholesterol biosynthesis. In addition, cholesterol and its metabolites have been involved in several oncogenic pathways, which echo uncontrolled cholesterol metabolism. The capacity of cholesterol for modulating cancer stem cells has stoked the discussion of cancer recurrence and drug resistance, while the immunomodulatory effect can contribute to promising immunotherapy.

Though insightful advancements have been made in cancer research, there are still questions to be answered. Are there any other hidden factors that, left unknown, contribute to the inconsistent results in epidemiological and clinical studies of cholesterol in cancer, despite the promising anti-tumor effects in laboratory data? What are the factors determining the preferential utilization of cholesterol between cancer cells and immune cells? How do they complete with each other in harvesting cholesterol as their utmost nutrient? Can any other cholesterol derivatives contribute to cancer development or perform immunosuppressive roles? Moreover, if cholesterol is understood as an essential compartment in cellular integrity, any artificial alterations in cholesterol metabolism could be compensated by the cell itself. Such alterations, similar to every medication, take statins as an example, could result in tremendous side effects, such as aching muscles and disruptions in liver and stomach functions. Could cholesterol homeostasis or other metabolisms be restored naturally without any drug interventions? By answering these questions, we shall gain more knowledge of the molecular mechanisms of cholesterol homeostasis and cancer development, which may potentially shine more light on cancer eradication.

## Figures and Tables

**Figure 1 cancers-12-01410-f001:**
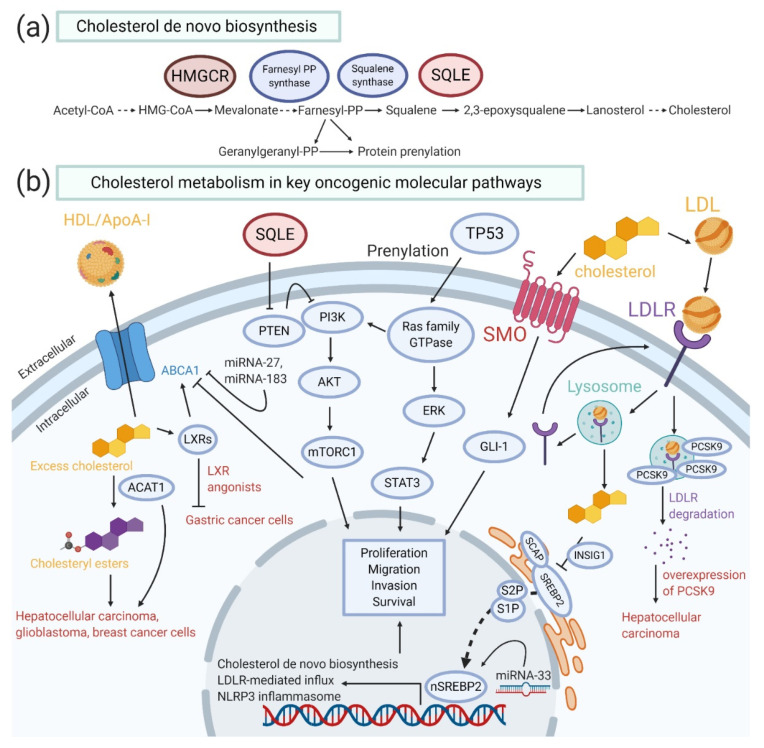
Cholesterol metabolism and key oncogenic pathways related to cancer development. (**a**) Cholesterol de novo biosynthesis. Starting from three molecules of acetyl-coenzyme A (CoA), cholesterol is synthesized in more than 20 enzymatic steps, whereas 3-hydroxy-3-methylglutaryl-CoA reductase (HMGCR) and squalene epoxidase (SQLE) act as rate-limiting enzymes. (**b**) Systematic diagram showing the cholesterol metabolism in relation to key oncogenic molecular pathways. Sterol-regulatory element binding protein 2 (SREBP2) regulates the transcriptional activity of cholesterol biosynthesis genes, low density lipoprotein receptor (LDLR)-mediated cholesterol influx, and Nod-like receptor protein 3 (NLRP3) inflammasome-associated inflammation. Embedded in SREBP2 gene, microRNA (miRNA)-33 can positively regulate SREBP2 expression. The over-activated cholesterol biosynthesis contributes to uncontrolled cell growth. Overexpressed proprotein-convertase-subtilisin-kexin type-9 (PCSK9) facilities the lysosomal degradation of LDLR, induces hypercholesterolemia, and eventually leads to the development of hepatocellular carcinoma. Excess cholesterol is exported via ATP-binding cassette (ABC) subfamily A member 1 (ABCA1) under liver X receptor (LXR) activation. However, in cancercells, ABCA1 is prohibited by the phosphoinositide 3-kinase (PI3K)/protein kinase B (Akt)/mammalian target of rapamycin complex 1 (mTORC1) pathway. The overall retention of intracellular cholesterol facilitates acyl-CoA:cholesteryl acyltransferase 1 (ACAT1), converting cholesterol into cholesteryl esters, leading to the development of different types of cancer. ABCA1 can also be inhibited by miRNA-27 and miRNA-183. TP53-mediated SREBP2 activation increases the production of farnesyl pyrophosphate (FPP) and geranylgeranyl pyrophosphate (GGPP) in the mevalonate pathway, resulting in prenylation of small Ras family GTPases and their downstream effectors. An increased SQLE level under high nuclear SREBP2 (nSREBP2) induction inhibits phosphatase and tensin homolog (PTEN) activity and sequentially allows the PI3K/Akt/mTORC1 signaling cascade. Lastly, cholesterol or its oxidative derivatives activate Smoothened receptor (SMO) in the sonic hedgehog (SHH) pathway. The overall alterations in these pathways increase the proliferation rate and the migration and invasion capacities, allow cell survival, and induce tumor formation.

**Table 1 cancers-12-01410-t001:** Clinical and epidemiological studies linking cholesterol, statin use, and cancer risks.

Year	Study Design	Population Group	Number	Main Findings	Reference
1967–1999	Prospective cohort study	Finland	>9000	High dietary cholesterol intake was associated with increased risk of colorectal cancer	[35]
1970–2007	Prospective cohort study	Scotland, United Kingdom	>12,000	Plasma cholesterol was positively related to risk of high-grade prostate cancer incidence	[36]
1972–2005	Cohort study	Norway, Austria, and Sweden	>500,000	Total serum cholesterol level was inversely associated with risk of overall cancer in females and with risk of liver, pancreas, and melanoma cancers in males	[37]
1972–2012	Prospective cohort study	Norway	>2000	An inverse association was found between cholesterol level and risk of prostate cancer	[5]
1984–2009	Randomized controlled trials	28 pharmacologic intervention arms and 23 control arms	>600,000	An inverse association was found between HDL-C level and risk of cancer incidence	[38]
1985–2002	Prospective cohort study	Finland	>20,000	No association between total cholesterol and risk of non-Hodgkin lymphoma (NHL), but an inverse association between HDL-C and NHL	[39]
1988–2002	Prospective study	Japan	>2000	An inverse correlation between serum cholesterol level and the incidence of gastric cancer	[6]
1990–2012	Meta-analysis	12 case-control studies and 4 cohort studies	>4000	Dietary cholesterol intake contributed to higher risk of pancreatic cancer	[40]
1992–2006	Prospective study	Korea	>1,000,000	High cholesterol was related to prostate, colon, and breast cancers and inversely related to lung, liver, and stomach cancers.	[41]
1995–2013	Case-control study	United Kingdom	>100,000	A decreased risk of colorectal cancer with statin use	[42]
1997–2001	Population-based case-control study	Shanghai, China	>800	Low HDL-C was related to higher risk of gallbladder and bile duct cancers. A U-shape relationship was found between total cholesterol level and LDL-C with biliary tract cancers	[43]
2002–2012	Retrospective study	Guangzhou, China	>600	Low HDL-C was correlated with poorer disease-free survival and overall survival in stage II/III colorectal cancer	[44]
2014	Meta-analysis	22 randomized controlled trials, 5 cohorts, and 6 case-control studies	>5,000,000	A significant risk reduction of liver cancer in all statin users, regardless of the type of statin used	[45]
1993–2011	Meta-analysis	15 cohort and 12 case-control studies	>1,000,000	A decreased risk of prostate cancer in statin users, though long-term statin use did not affect the total risk of prostate cancer	[46]
2012	Meta-analysis	8 observational and 3 post-hoc analyses of 26 clinical trials	>5000	A significant drop of over 30% in gastric cancer with statin use; significance remains in both Asian and Western populations	[47]
1998–2014	Retrospective case-control study	United States	>400,000	A drop of over 60% risk in pancreatic cancer with statin use of more than 6 months	[48]

**Table 2 cancers-12-01410-t002:** Therapeutic targets of cholesterol homeostasis.

	Chemical	Inhibitory Target	Effects	Reference
Cholesterol biosynthesis	Statins	HMGCR	Block the formation of mevalonate from HMG-CoA	[137,154]
	δ-, γ-, and α-tocotrienol			[125]
	β-ionone			[126]
	geranylgeraniol			[127]
	geraniol			[128]
	S-allylcysteine			[134]
	Bisphosphonate	FPP synthase	Prevent the prenylation of small GTPases	[155]
	Lapaquistat	Squalene synthase	Block the conversion from FPP to squalene	[156]
	Zaragozic acid			[157]
	RO 48-8071	OSC	Block the synthesis of lanosterol from 2,3-monoepoxysqualene	[158,159]
	Curcumin	SQLE	Block the synthesis of squalene epoxide	[135]
	Garlic extract	Sterol 4α-methyl oxidase	Prevent the formation of zymosterol	[133]
Cholesterol intake	Ezetimibe	NPC1L1	Reduce LDL-C levels	[160]
	Curcumin			[136]
	EGCG	LDLR		[130,131,132]
Cholesterol esterification	Avasimibe	ACAT1	Reduce the formation of cholesteryl esters	[161]
Cholesterol depletion agent	Methyl-β-cyclodextrin	Lipid rafts	Facilitate the depletion of cholesterol from membranes	[162]
Immunotherapy	Dendritic cell vaccine		Increase antigen presentation efficiency	[121]

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
