# Peer review of "The Pivotal Role of the Dysregulation of Cholesterol Homeostasis in Cancer: Implications for Therapeutic Targets"

_cancers, 2020, doi:10.3390/cancers12061410_

Round 1
Reviewer 1 Report
The manuscript entitled “The pivotal role of the dysregulation of cholesterol homeostasis in cancer: Implications for therapeutic targets” by Mok and Lee presents timely and recently reported important information, e.g., that E3 ubiquitin ligase MARCH6 is recruited to degrade squalene epoxidase when excess cholesterol is present, that ACAT1 elevation is revealed under proteomic and phospho-proteomic analyses in HCC, etc, while highlighting past and more recent important information on the contradictory roles of cholesterol in different cancers and cell types, including different types of immune cells, as well as briefly discussing therapeutic targets of cholesterol homeostasis in the context of cancer.
Minor issues:
- Line 29: Instead of “contractive”, was “contradictory” meant?
- Line 39: Please clarify how cholesterol is involved in “compositing”.
- Figure 1: “Prolifeartion” should be corrected to “Proliferation”.
- Line 60: Instead of “light-density”, was “low density” meant? Please also check through entire manuscript for mention of “light-density” instead of “low density”.
- Line 108-109: Instead of “total serum cholesterol (TSC)”, perhaps better to use “total cholesterol (TC)”, given that some assays may have been done on plasma as well as serum samples. Please also check through entire manuscript for mention of “TSC”.
- Line 268-270: Regarding “In the tumor microenvironment, high cholesterol levels can lead to CD8+ T cell while inducing immune checkpoints”, please clarify what is meant by “high cholesterol levels can lead to CD8+ T cell”.
Author Response
Please kindly see the attachment".
Thank you!

Reviewer 2 Report
In this review article, the authors provided a quite exhaustive discussion of the literatura data linking alterations in cholesterol homeostasis and cancer cell physiopathology.
The collected findings highlight that high cholesterol dysregulations are often associated to the increased risk of developing cancer.
From a molecular/cellular point of view, a number of cancer cells are characterized by a lively cholesterol metabolism, which is found to be upregulated. In addition, the activation cholesterol biosynthesis is strongly associated to cancer stem cell proliferation.
Cholesterol also plays a pivotal roles in regulating several processes involving immune cells, thus potentially altering the response of the immune system in the tumor microenvironment.
When evaluated as a whole, this manuscript appears well-written and well organized.
Despite the novelty is partially undermined by a very recent published article (Huang et al., 2020. Cholesterol metabolism in cancer: mechanisms and therapeutic opportunities, Nature Metabolism), I believe this is a hot topic, thus the work presented by Mok and Lee may have some merit and can meet the interests of the readers of cancers.
Some concerns:
The whole manuscript is well written, however it should be carefully edited as style and typographical errors are present.
The authors clearly highlighted that cholesterol requirement is higher in a number of tumoral contexts. Coherently with this notion, the authors also summarize several studies aimed at demonstrating the potential anti-tumoral effect of cholesterol lowering drugs, alone or in combination with classic antineoplastic agents. In my opinion, the manuscript will acquire more strength by adding information about the effect of drugs, commonly known for their anticancer properties, in reducing cell cholesterol. Notably, increasing evidence suggests that anticancer drugs may exert their antiproliferative activities at least in part by reducing cholesterol content/biosynthesis. For instance, it has been demonstrated that doxorubicin induces cancer cell death by decreasing HMGCR expression and reducing cholesterol levels (Yun et al., 2018, Anti-cancer effect of doxorubicin is mediated by downregulation of HMG-Co A reductase via inhibition of EGFR/Src pathway, Lab Invest). Other studies indicated that Tamoxifen modulates cholesterol metabolism (Segala et al., 2013, 5,6-Epoxy-cholesterols Contribute to the Anticancer Pharmacology of Tamoxifen in Breast Cancer Cells, Biochem Pharmacol). In addition, a recent article highlighted that the anti-cancer molecule JQ1 severely impacts the expression of proteins involved in cholesterol metabolism, thus leading to a strong decrease of cholesterol content in the human liver cancer cell line HepG2. Interestingly, the same report shows that the administration of cholesterol counteracts the anti-proliferative effect induced by JQ1 in liver cancer cells, and the acquisition of JQ1-resistance is accompanied by a compensatory upregulation of proteins belonging to cholesterol homeostasis (Tonini et al., 2020. Inhibition of Bromodomain and Extraterminal Domain (BET) Proteins by JQ1 Unravels a Novel Epigenetic Modulation to Control Lipid Homeostasis. Int J Mol Sci).
These findings should be integrated and discussed into the manuscript. The fact that anti-neoplastic drugs may reduce cell proliferation (at least partially) by affecting cholesterol metabolism per se, further strengthen the importance of this metabolic pathway in the field of oncology.
Reviewer 3 Report
In the manuscript entitled "The pivotal role of the dysregulation of cholesterol homeostasis in cancer: Implications for therapeutic targets" by Etienne Ho Kit Mok and Terence Kin Wah LEE authors review the literature on the de-regulation of cholesterol homeostasis in cancer discussing the treatment options developed to date.
The topic is of considerable relevance but in my opinion, the manuscript can be improved by reviewing the structure of some paragraphs and providing other missing information. Below are my comments:
-From the title of paragraph 2, I would expect the discussion of the work on the de-regulation of cholesterol homeostasis in cancer cells. Instead, the paragraph is generally a description of the mechanisms of cholesterol regulation, the cancer part is marginal in this section. I suggest therefore changing the title of paragraph 2 and leaving it as an introduction to the mechanisms of cholesterol regulation and to include in this paragraph Figure 1, Part A;
- Paragraph 4 is the one discussing the deregulation of cholesterol homeostasis in cancer; Figure 1B should be included in this paragraph and also some information reported in paragraph 2.
- In paragraph 4, on the other hand, the pathways to deregulation of cholesterol in cancer are discussed. This, in my opinion, is the central paragraph of the review that needs to be improved by adding missing information on relevant pathways involved in cholesterol deregulation in cancer. For example, the correlation between cholesterol/inflammation-inflammasome/cancer should be included as well as some microRNAs that regulate cholesterol metabolism in tumors.
- Increasing evidence shows that many compounds of natural origin can be used to regulate cholesterol homeostasis not as primary therapy but as adjuvant or preventive agents. I believe that in section 7 some of these studies should be mentioned and discussed.
-Check and format the text in the correct way in paragraph 7 line 338-343 and 353.
Author Response
"Please see the attachment".
Thank you!

Round 2
Reviewer 2 Report
The authors provided an exhaustive point-by-point response letter, and properly addressed all the reviewer's concerns.
Reviewer 3 Report
the authors have provided a comprehensive response to the points I raised.